# Diabetes Mellitus and Hypertension—A Case of Sugar and Salt?

**DOI:** 10.3390/ijms21155200

**Published:** 2020-07-22

**Authors:** Marcus Sondermann, Michał Holecki, Andrea Marita Kirsch, Manuela Bastian, Dagmar-Christiane Fischer, Holger Sven Willenberg

**Affiliations:** 1Division of Endocrinology and Metabolism, Rostock University Medical Center, 18057 Rostock, Germany; m.sondermann@klinikum-brandenburg.de (M.S.); kirsch_andre@gmx.net (A.M.K.); 2Department of Internal, Autoimmune and Metabolic Diseases, Medical Faculty in Katowice, Medical University of Silesia, 40-055 Katowice, Poland; holomed@gmail.com; 3Institute for Clinical Chemistry and Laboratory Medicine, Rostock University Medical Center, 18057 Rostock, Germany; manuela.bastian@med.uni-rostock.de; 4Department of Pediatrics, Rostock University Medical Centre, Rostock, 18057 Rostock, Germany; dagmar-christiane.fischer@med.uni-rostock.de

**Keywords:** diabetes mellitus, hypertension, adrenal, glucose, sodium, osmolality, cortisol, aldosterone, SGLT2, water homeostasis

## Abstract

The majority of patients with diabetes mellitus (DM) have hypertension (HTN). A specific mechanism for the development of HTN in DM has not been described. In the Zucker, Endothel, und Salz (sugar, endothelium, and salt) study (ZEuS), indices of glucose metabolism and of volume regulation are recorded. An analysis of these parameters shows that glucose concentrations interfere with plasma osmolality and that changes in glycemic control have a significant impact on fluid status and blood pressure. The results of this study are discussed against the background of the striking similarities between the regulation of sugar and salt blood concentrations, introducing the view that DM is probably a sodium-retention disorder that leads to a state of hypervolemia.

## 1. Introduction

There are multiple conditions and risk factors leading to a chronic increase in blood glucose concentrations. As such, diabetes mellitus (DM) is a heterogeneous group of disorders [1] and in more than 60% of patients with DM, hypertension (HTN) is present as well [2]. DM is more prevalent in the elderly and comes with a typical pattern of clinical and laboratory features at the time of diagnosis [3]. Notably, people above 55 years of age have a 90% chance of developing HTN, while patients with DM experience such a high prevalence of HTN approximately 10 years earlier [4]. A specific mechanism for the development of HTN in DM has not been found and a number of factors have been addressed so far, including overactivity of the sympathetic nervous and the renin–angiotensin–aldosterone systems (RAAS), aberrant renal sodium handling, endothelial dysfunction, blood vessel damage, insulin resistance, chronic inflammation, and activation of adrenal steroid hormone secretion [5,6,7,8].

However, principal considerations on the regulation of glucose and salt plasma concentrations show striking similarities and suggest the view that DM is probably a sodium-retention disorder. As sketched in Figure 1, blood concentrations of glucose and sodium are also increased via hormonal mechanisms involving the adrenal gland and secretion of glucocorticoids and/or mineralocorticoids and bind to their respective receptors. Glucose and sodium bind water, raise the osmotic pressure, and stimulate release of vasopressin, which leads to thirst and reabsorption of water in the kidney. Hypervolemia develops, disguising the total body content of glucose and sodium, and promotes the development of hypertension for which vasopressin concentrations may become inadequately high.

In Figure 1, the thoughts for such an assumption are further developed in additional aspects and substantiated with results collected within the frame of the Zucker, Endothel, und Salz (sugar, endothelium, and salt) study (ZEuS).

## 2. Results

Within the frame of the ZEuS study, data on plasma osmolality in relation to the results of an oral 75 g glucose challenge (arm A, ZEuS/oral glucose tolerance test (oGTT)) or indices of glucose metabolism and blood pressure (arm B, ZEuS/DM) were obtained (Table 1). All hypertensive patients were on antihypertensive treatment, including beta-blockade (43.3%), anti-angiotensin agents (38.3%), diuretics (30.5%), or calcium channel blockers (20.3%).

Out of 73 individuals undergoing an oral 75 g glucose challenge test, 19 (26%) were diagnosed with impaired glucose tolerance and four (5.5%) with DM. Interestingly, insulin and homeostasis model assessment-insulin resistance (HOMA-IR) values were normally distributed (15.9 ± 11.2 µU/mL and 3.7 ± 2.5, respectively), and correlated well with systolic blood pressure (Person-*r* = 0.753/*p* < 0.05 and 0.816/*p* < 0.01, respectively) and with plasma osmolality (Person-*r* = 0.719/*p* < 0.01).

Notably, 20 (27%) had plasma osmolality values higher than 300 mosmol/kg, whereby half of them were in the group of individuals with elevated glucose concentrations after the oral glucose challenge (*p* < 0.05 in the Chi-square test). There was also a relevant correlation between plasma osmolality and serum copeptin concentrations (Spearman-*r* = 0.321, *p* < 0.05) with a steep increase in copeptin concentrations for plasma osmolality values of more than 300 mosmol/kg (Figure 2, Panel a).

Of the 24 patients with DM—all on intensified insulin treatment, including three with an insulin pump—more than 78% had type 2 and 21% had type 1 diabetes, whereby patients with type 1 diabetes were obese and regarded as having double diabetes. As expected, comorbidities were present in a number of patients and distributed as follows: peripheral polyneuropathy in 47.8%, nicotine in 39.1%, chronic kidney injury in 13.0%, glaucoma in 4.3%, and cardiovascular disease in 65%, with parental hypertension in 69.9%. Please see Table 1 for further details.

In these individuals, plasma osmolality showed a significant correlation with systolic (Spearman-*r* = 0.402, *p* = 0.05), but not with diastolic blood pressure. Systolic blood pressure did not change significantly during the study (SPB 128.6 ± 11.7 vs. 131.2 ± 19.5 mmHg), and diastolic blood pressure dropped slightly and scarcely failed significance (77.9 ± 8.0 vs. 71.2 ± 7.4 mmHg). In this group of patients with DM, education and changes in insulin treatment led to the expected drop in hemoglobin A1c (HbA1c) (8.40 [7.65–9.05] % vs. 7.25 [6.40–8.05] %; *p* < 0.001). In 10 of these patients, data on serum N-terminal brain natriuretic peptide (NT-proBNP) concentrations obtained prior and after improved glucose metabolism by the modified insulin therapy revealed a nonsignificant increase of NT-proBNP, in parallel to the intensification of insulin therapy (102 [59.2–260.2] vs. 123 [57.8–278.5] ng/mL). Concentrations of the NT-proBNP, however, were lower the more HbA1c dropped after intervention (Figure 2, Panels b–d). Changes in renin (17.0 [4.2–37.2] vs. 15.8 [8.1–85.4] pg/mL) and aldosterone concentrations (104 [81–231] vs. 149 [75–282] pmol/L) were in favor of relief from volume stress but did not, however, reach statistical significance (the distribution was too wide for the number of patients).

## 3. Discussion

The majority of patients with DM develop HTN. On one hand, HTN in patients with DM is thought to arise from common risk factors related to the development of DM, e.g., genetic disposition, overactivity of the sympathetic nervous system, chronic inflammation, and excess adrenal steroid hormone secretion, along with RAAS activation as a consequence of surplus adipose tissue and a long list of factors generated by these fat cells [4,5,6,7,8,10]. On the other hand, HTN is also discussed to be already a consequence of diabetes with renal dysfunction, RAAS activation and abnormal renal sodium handling, endothelial dysfunction, blood vessel damage, and chronic inflammation [4,5,6,9]. A certain degree of vagueness may be the reason that until now, DM has not been considered a cause of secondary HTN [11,12].

However, multiple studies have shown that a salt-mediated increase in osmolality triggers the release of arginine-vasopressin [13,14]. Our data on osmolality and copeptin reflect this relationship and the dose-response curve is similar to the one reviewed recently [15]. Interestingly, it is known that acute effects of salt administration on blood pressure are explained by an increase in osmolality [16]. Likewise, vasopressin-deficient Brattleboro rats do not develop hypertension when fed salt and sodium-retaining agents before active vasopressin is administered [10,17,18,19,20]. Similar to the results of our study, the relation of osmolality and blood pressure is also a chronic one. From a physiological and historical point of view, sugar is as much a valuable crystal as is sodium chloride. While salt was once also called “white gold”, 80% of slaves in the Caribbean worked for sugar cane farmers [21]. Both molecules are most likely tasty for evolutionary reasons and extensively used today by the food industry [22]. They bind water, are up-regulated by adrenal gland hormones (sodium by mineralocorticoids, sugar by glucocorticoids), which are synthesized by different enzymes that evolved by gene duplication (aldosterone synthase, *CYP11B2,* and 11β-hydroxylase, CYP11B1), and they bind to closely related receptors (mineralocorticoid or glucocorticoid receptors, respectively) after inactivation into cortisone or reactivation into cortisol by 11β-hydroxysteroiddehydrogenase type II or type I action, respectively (Figure 1).

However, an important aspect is that sodium and glucose are reclaimed from the kidney filtrate through sodium/glucose cotransporters (SGLT). Thus, as a result of the renal threshold and continuous SGLT action, overfeeding with sugar and hyperglycemia would induce a perpetual recycling process of glucose leading to an increase in total body glucose and to sodium retention. As pointed out in more detail elsewhere, the retention of sodium and glucose results in a state of hypervolemia that develops through induction of thirst and the action of vasopressin, and is reflected by blood pressure, copeptin, and NT-proBNP levels [10,23]. Within this context, it is noteworthy that NT-proBNP concentrations indeed dropped in patients with steep decreases in HbA1c values. Diastolic blood pressure values also declined after the intervention in glucose metabolism, however, they just missed statistical significance. A state of hypervolemia may increase cardiac preload and cause shear stress in vessels, thus promoting chronic heart failure and conditions with inflammation in patients with diabetes or hyperaldosteronism. All in all, this would also explain the excellent cardiovascular outcome of patients in studies with SGLT-2 inhibitors [24] and is not contradicted by the decrease in urinary sodium excretion in patients on long-term treatment with such agents, as the latter resembles a physiological aldosterone escape phenomenon [10].

Problems in volume regulation may also come with the progression of chronic kidney disease (CKD). Thus, our results may change at different stages of disturbed glucose metabolism or in the presence of glucosuria, as well as CKD stages in combination with electrolyte imbalances and sensitivity to aldosterone. Notably, an octet of changes were found to underlie the development of type 2 diabetes mellitus, whereby the increase in glucose reabsorption is one feature only, and therefore, explains the development of hypertension only partially [25,26]. Our findings may also be altered by interference with volume changes in the treatment of CKD or provoked by hyperfiltration in different stages of hypertension. Notably, any therapeutic intervention into the renin-angiotensin-aldosterone system may have an effect that outbalances the salt-retention provoked by reabsorption of glucose and could change our findings. However, because of this, therapeutic interventions to inhibit the activity of the renin-angiotensin-aldosterone system and to lower salt and volume load may be especially effective in the therapy of CKD in individuals with diabetes. Targeting diabetic kidney disease will significantly reduce morbidity, and eventually, mortality. Notably, promising agents have been shown to interfere with inflammatory and hemodynamic pathways [27]. As such, atrasentan and SGLT2 inhibitors decrease albuminuria as well as systolic and diastolic blood pressure indices [27].

However, we noticed a nonsignificant slight elevation of NT-proBNP concentrations with intensified insulin treatment in a portion of our patients. This might explain the hitherto noted weight gain with insulin treatment, and at least in part, the fact that insulin treatment has a U-shaped curve regarding survival [28]. On the basis of these findings and our observations, it might be an option to combine insulin therapy with oral glucose-lowering agents, such as SGLT-2 inhibitors, to improve the outcome.

## 4. Patients and Methods

In the ZEuS study, the relation of parameters reflecting the glucose metabolism were studied along with indices of salt, water, and volume homeostasis. The study was approved by the institutional review boards and local ethics committees and informed consent was obtained from all individual participants included in the study.

In arm A (ZEuS/oGTT), adult individuals undergoing an oral 75 g glucose load were investigated for anthropometric measures, glucose, insulin, osmolality, and copeptin concentrations at baseline, after 60 as well as 120 min. The main reason was to look for diabetes or other abnormalities in glucose metabolism as well as for the presence of insulin resistance in patients with weight gain or obesity. This program was established simultaneously at the Rostock University Medical Center in Germany and the Medical University of Silesia in Poland. Plasma samples scheduled for determination of copeptin were stored at −80 °C until analysis. The Kryptor system (B.R.A.H.M.S./ThermoFisher Scientific) was used for copeptin measurements and the Osmomat 030 freezing point osmometer (Gonotec, Berlin) for osmolality determinations throughout the study—they were all performed at the Institute for Clinical Chemistry and Laboratory Medicine, Rostock University Medical Center.

In arm B (ZEuS/DM), patients with DM were investigated for anthropometric measures, multiple parameters of glucose metabolism, including HbA1c (photometry), insulin, or C-peptide (both ECLIA Cobas e411, Roche, Mannheim, Germany), and indices of salt-water homeostasis, including renin (Liaison platform, DiaSorin, Saluggia, Italy), aldosterone (Chromogene LC-MS/MS assay), NT-proBNP (Cobas ECLIA), blood pressure, and volume status. In this study arm, patients were included who were in need of specific education and changes in their insulin treatment regiments, e.g., a switch from a conservative to a more intensified form of insulin injection therapy, or from multiple insulin injections to therapy with an insulin pump. Patients with polyuria did not qualify for this study. All in all, 24 adult patients who were capable of full self-management of their diabetes were included and studied before and re-evaluated after 3–6 months of change in therapy. It is important to mention that the medication not changing during the ZEuS/DM study was an exclusion criterion for follow-up in arm B. Furthermore, medically-related forms of diabetes were not included. The aim was to ensure a lifestyle of rather high caloric intake, which was evaluated beforehand.

Methodological limitations are to be seen in the number of patients in the follow-up study, which was caused in part by the exclusion of patients who experienced a change in the blood pressure modifying medication—irrespective of the fact that negative assertions were made on larger case numbers. We also studied indices of glucose and sodium concentrations and not actually total body glucose, total body sodium, or intravascular volume. Apart from these issues, our conclusions therefore maintain a certain degree of interpretation of our results, rather than direct evidence. Furthermore, the osmotic properties of glucose cannot be discerned from the osmotic properties of salt along with the regulatory consequences, when both systems are interrelated as described above. Nevertheless, this close connection should be kept in mind when reading other study results. Since the number of diabetic probands was rather manageable and since we sought to include patients with type 2 DM and/or obesity, we could not differentiate between diabetes types. Therefore, it may well be that our findings cannot be translated to individuals with other forms of DM.

## 5. Conclusions

With the abovementioned limitations in mind, we conclude that the regulation of glucose concentrations by cortisol has many aspects in common with the regulation of sodium concentrations by aldosterone. Both sugar and salt regulate blood osmolality, which triggers thirst as well as copeptin/vasopressin release and which is related to blood pressure. Through reabsorption of sugar and salt in the kidney, elevations in plasma glucose concentrations promote sodium retention and elevations in NT-proBNP, most likely by fluid load. Thus, DM is probably a sodium retention disorder with all its negative consequences.

## Figures and Tables

**Figure 1 ijms-21-05200-f001:**
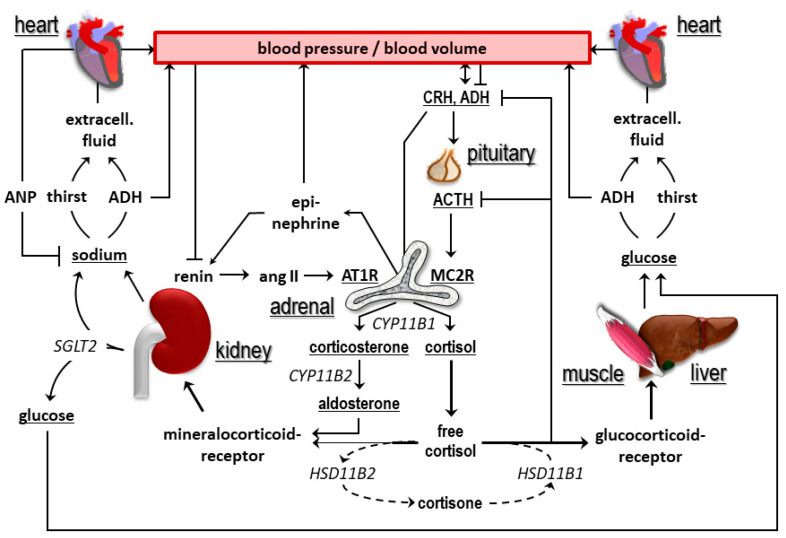
Parallelisms in the regulation of glucose and salt blood concentrations. This figure is a further development of the scheme in [9]. Notably, both glucose and sodium are reclaimed in the kidney through the action of sodium-glucose cotransporters.

**Figure 2 ijms-21-05200-f002:**
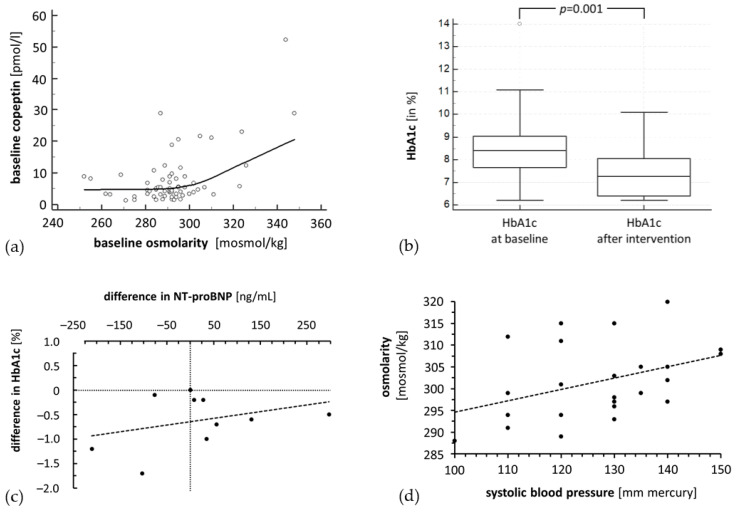
The ZEuS/oGTT study (arm A) showed that there is a significant correlation between baseline osmolality and copeptin concentrations (**a**). Patient education and changes in insulin treatment regiments reduced HbA1c values significantly in the ZEuS/DM study (arm B) of 24 patients with diabetes mellitus (**b**). In 10 of these patients, serum NT-proBNP concentrations were available and shown to rather increase with intensification of insulin therapy but remained lower the more HbA1c values dropped (**c**), negative figures indicate a smaller difference between pre- and postinterventional values. Plasma osmolality correlated with systolic blood pressure values (**d**).

**Table 1 ijms-21-05200-t001:** Baseline characterization of individuals in the two arms of the ZEuS study. While the portion of female patients was higher in arm A intensivation of insulin therapy was more frequently done in male patients (arm B). The patients in arm A were younger and suffered less frequently from hypertension. Their BMI, however, was higher.

Parameter	Age	Gender	BMI	SBP	DBP	Glucose	HbA1c	HTN
(unit)	(years)	(% female)	(kg/m^2^)	(mmHg)	(mmHg)	(mmol/L)	(%)	(%)
ZEuS/oGTT (arm A, *n* = 73)	49.1 ± 18.0	71.6	32.2 ± 7.7	131.4 ± 13.6	78.8 ± 10.3	5.2 ± 0.6	5.5 ± 0.4	58.3
ZEuS/DM (arm B, *n* = 24)	57.8 ± 16.0	20.8	29.8 ± 8.2	128.6 ± 11.7	77.9 ± 8.0	12.1 ± 4.3	9.2 ± 2.7	65.7

Abbreviations: body mass index (BMI), diastolic blood pressure (DBP), diabetes mellitus (DM), hypertension (HTN), oral glucose tolerance test (oGTT), systolic blood pressure (SBP), Zucker, Endothel, und Salz (sugar, endothelium, and salt) study (ZEuS).

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
