# Peer review of "Diabetes Mellitus and Hypertension—A Case of Sugar and Salt?"

_ijms, 2020, doi:10.3390/ijms21155200_

Round 1

Reviewer 1 Report

I recommend expanding the possible therapeutic consequences as indicated in the Lacava review in the section "Molecules active on hemodynamic pathways"

(Lacava et al Novel avenues for treating diabetic nephropathy: new investigational drugs. Expert Opin Investig Drugs. 2017 Apr;26(4):445-462.)

Author Response

Dear Madam and Sir,

dear Professor Fischer,

thank you very much for reviewing our manuscript: »Diabetes mellitus and hypertension – a case of sugar and salt?«

We found the comments of both reviewers very helpful to improve our manuscript. Please, find our detailed response below.

Reviewer 1

We have modified our discussion as proposed and cited the paper of Lacava et al. 2017.

Reviewer 2 Report

The work of Sondermann et al focuses on the relationship between hypertension and DM and underlying mechanisms. For this purpose, the authors carry out a study with two arms: one (A) evaluating people (n = 73) with and without diabetes, studied with an OGTT; other (B) considering people with diabetes treated with insulin (n = 24). Glucose metabolism and blood pressure parameters are evaluated in both arms.

This is a relevant issue considering that the knowledge of mechanisms underlying this relationship can have an important therapeutic impact. Lately, renal metabolism of salt, glucose and insulin has been extensively reviewed, and knowledge in this area has advanced, greatly motivated by the results obtained with the use of SGLT2 in the treatment of DM2. Nonetheless, I have major concerns better explained in the following paragraphs.

General comments:

Despite the interest and topicality of the subject, the manuscript lacks relevant methodological and results details to allow a thorough interpretation. Overall it is difficult to make more specific comments on the text, thus I will mainly comment by section.

Introduction

In the Introduction section, the authors write about the relationship between DM and HTN. However, they do not clarify what type of diabetes they refer to. There are well known differences in pathophysiology of DM types (namely Type 1 and Type 2) and in their relationship with hypertension. This should be considered, and the study framed in its context, underlining differences that may impact the results.

Results

Results section must be completed with the relevant information for the study, in order to allow its interpretation and review.

Baseline characterization of individuals is lacking, namely age, gender, pathology, more specifically, hypertension, diabetes (type of), and other co-morbidities.  Blood glucose baseline values, particularly in individuals with decompensated diabetes in arm B should be addressed. Regarding the diagnosis of diabetes, and given the potential differences in pathophysiology, the authors should also include the characterization of diabetes in insulin-treated patients in arm B. Additionally, authors do not mention baseline therapy for individuals. Considering the subject under study, it seems relevant to indicate the use of antihypertensive drugs, and their class when present. The authors also refer to anthropometric measures and those related to glucose and salt metabolism in the methodology; however, most of them are not reported in the results section.
For Arm A and B there are no indication to blood pressure diagnosis and antihypertensive medication. In Arm A, the author should report blood pressure measurements and changes during the study. In arm B, authors do not report changes in osmolality and blood pressure after the intervention. The results obtained for baseline renin are also unclear. The NT-proBNP, insulin and HbA1c relationship should be clarified.

Discussion

Authors should discuss the potential differences in the relationship between hypertension and DM according to the type of diabetes. Authors also mention that the results of osmolality and copeptin reflect the fact that an increase in salt-mediated osmolality leads to an increase in vasopressin. However, it was not demonstrated by the authors that the increase in osmolality in the individuals studied in arm A or in people with decompensated DM included in arm B is related to salt, nor was it demonstrated in the latter a relationship with copeptin. Patients included in arm B can have several causes for increased osmolarity, including hyperglycemia and polyuria resulting from osmotic diuresis.Regarding NT-proBNP, as explained by the authors, it increased after the intensification of therapy and the reduction of HbA1c (Fig1c). In the legend, the authors refer that a negative number indicates a smaller difference. What does a difference of -250 means? Finally, the authors should point out limitations of their work. 

Methodology

The Methodology section should be improved and clarified. It is not clear the study design, inclusion, exclusion criteria, and the number of people included in each arm. The authors also do not describe how they performed parameter measurements, namely regarding techniques or kits used. There are parameters referred to in the Methodology section that are not reported in the Results section, as well as the other way around, e.g.  serum copeptin measurement technique is not described in the Methodology section.

Conclusion

Finally, considering the submitted manuscript I do believe it is not possible to draw the conclusions mentioned by the authors, namely regarding to salt as the etiological factor of the findings.

Minor comments:

Figure 1 panels include plots on both arms. I suggest that the authors clarify that, or separate results from both arms in different figures. Figure 2 must also be described and clarified, within text context.

Author Response

Dear Madam and Sir,

dear Professor Fischer,

thank you very much for reviewing our manuscript: »Diabetes mellitus and hypertension – a case of sugar and salt?«

We found the comments of both reviewers very helpful to improve our manuscript. Please, find our detailed response below.

Reviewer 2

General comments

            We have expanded our Methods and our Results sections.

Introduction

Diabetes mellitus may develop for several reasons. There is a long list of risk factors, including lack of motion/walking, abundance of calories in relation to degradation, usage of interfering medication, certain genetic polymorphisms, alcohol consumption / history of pancreatitis, liver disease, activation of the hypothalamic pituitary axis or the sympathoadrenal system for many reasons, and many others. We find it difficult to ascribe the diabetes type to many of our patients. This problem of the classical differentiation of diabetes types is reflected by work of Ahlqvist et al. (Lancet Endocrinol Diab 2017).

            However, in order to characterize the relation of diabetes and blood pressure, we seeked to include patients who were overweight or had a clear classification of type 2 diabetes mellitus for the assumed surplus in glucose consumption in relation to (physical) activity. We have studied patients with weight gain or obesity (arm A) or who had type 2 diabetes mellitus or type one with obesity (arm B). We made this more clear in the manuscript. This was an important point.

            We also introduced a paragraph describing the content of Figure 1 in more detail.

Results

            As stated by the Reviewer baseline characterization of the studied individuals is a requirement for interpretation of results. Therefore, we expanded this section of the manuscript by additional sentences dedicated on hypertention and treatment and provided a Table 1. In addition, we reported hormonal measurements as requested.

Discussion / Conclusion

            Since the osmotic properties of glucose can not be discerned from the osmotic properties of salt along with changes in the interrelated regulatory systems we are indeed not able to "prove" that sodium is the (sole) source of the changes described. In our revised manuscript, we expanded on this problem but made the point that the results of other studies are to be seen against the background of the issues raised by our findings and our work here.

            We are also well aware that different qualities of decompensations of glucose metabolism do have profound impact on fluid homeostasis. We, therefore, excluded patients with polyuria and/or exsiccosis. This was made more clear in the Methodology section and in the discussion, a paragraph on the limitations of this study was introduced in the Methodology section.

Methodology

            This section was also expanded for inclusion of assay information. A paragraph on the limitations of this study was introduced.

Altogether, we hope that our manuscript has improved and reached a quality which makes it more intersting for publication in the International Journal of Molecular Sciences.

Best regards,

Holger Willenberg

Round 2

Reviewer 2 Report

I sincerely appreciate the revisions and comments made by Sondermann et al., that add clarity to the manuscript. Diabetes heterogeneity (OMS, 1965; Li Li et al, 2015; Ahlqvist et al 2017; McCarty, 2017) relates also with diverse pathophysiological mechanisms. Therefore, population characterization is of utmost importance for the work interpretation.

Minor comments:

1) Introduction, line 39/40 - Do the authors really mean “upregulation” of glucose and sodium?

2) Considering that the authors refer the measurement of insulin/c-peptide it would be clearer (considering diabetes heterogeneity) if they could also include them in the results section as well as an insulin resistance parameter

3) It would be worth do discuss the impact/bias that the presence of CKD patients and patients under different antihypertensive medication in the studied sample can have on the results.

4) It may be worthwhile to show the kidney role in glucose reabsorption in Figure 1, in order to make it clearer to the reader.

5) The number of subjects in each arm (N) should be included in Table 1.

Author Response

Dear Madam and Sir,
dear Professor Fischer,

thank you very much for once again re-evaluating our manuscript: »Diabetes mellitus and hypertension – a case of sugar and salt?«

We found the comments of the reviewer again helpful. We used his ideas and reworked our manuscript, adding the data when possible. Please, find our detailed response below.

general comments
We took on the idea in the discussion of limitations paragraph and cited some of the provided references. Thank you.

minor comment 1)
Yes, glucocorticoids and mineralocorticoids (and precursors) upregulate sugar and salt. To avoid misunderstandings, we now opted for the word »increase«.

minor comment 2)
Insulin measurements were available for the ZEuS / oGTT study (arm A). We provided the data on concentration and the HOMA-IR index and its correlation with the systolic blood pressure in the Results section.
For the ZEuS / DM study the picure about insulin and/or C-peptide concentrations is far from complete. To avoid confusion, we did not report on the available results.

minor comment 3)
Done!

minor comment 4)
We have included the role of SGLT2 into the scetch and modified the cartoon.

minor comment 5)
We have included the patient numbers in Table 1.

Altogether, we hope that our manuscript has improved and reached a quality which makes it more intersting for publication in the International Journal of Molecular Sciences.

Best regards,
Holger Willenberg
